# Defining a New Classification System for the Surgical Management of Neuroendocrine Tumor Liver Metastases

**DOI:** 10.3390/jcm12072456

**Published:** 2023-03-23

**Authors:** Kelly M. Mahuron, Gagandeep Singh

**Affiliations:** Department of Surgery, City of Hope National Medical Center, Duarte, CA 91010, USA

**Keywords:** neuroendocrine liver metastases (NETLM), hepatic cytoreduction, hepatic debulking

## Abstract

Although rarely curative, hepatic cytoreduction of neuroendocrine tumor liver metastases (NETLM) is associated with improved symptom control and prolonged survival. Preoperative 68Ga DOTATATE and gadoxetic acid-enhanced liver MRI can improve characterization of hepatic disease extent to improve surgical clearance, and resection of the primary tumor is associated with improved survival regardless of whether the liver metastases are treated. As parenchymal-sparing surgical techniques and the lowering of the debulking threshold have expanded the numbers of eligible NETLM patients for hepatic cytoreduction, we propose a new classification system to help guide surgical management. A multimodal approach that includes surgery, liver-directed therapies, and systemic therapies has improved outcomes and increased longevity for patients with well-differentiated metastatic NET.

## 1. Introduction

Although rare, gastroenteropancreatic neuroendocrine tumors (GEP-NETs) are a heterogeneous group of typically slow-growing tumors with a rising incidence [1,2,3,4]. These tumors are classified based upon their site of origin, tumor grade, and cellular differentiation, and whether they secrete functional hormones [5]. While the majority of these tumors exhibit an indolent nature, they commonly present with advance disease, and up to 75% of patients develop liver metastases (LM) [6]. In approximately 50% of patients, the liver is the only site of metastatic disease [7], and hepatic tumor burden is often the main determinant of a patient’s prognosis [8]. Therefore, a multimodal treatment strategy utilizing surgery, liver-directed therapies, and systemic therapies has been adopted in the management of NETLM.

The gold standard for curative intent management of NETLM is surgery, and surgical candidacy largely depends on the burden of a patient’s hepatic disease. The majority of patients with NETLM tend to have multifocal hepatic involvement, and although complete resection is often not feasible, surgical debulking with cytoreductive surgery has been shown to be beneficial for symptom control and for improved overall survival [9]. Additionally, surgical resection has been reserved for patients with well-differentiated NETs, as poorly differentiated tumors, classified as neuroendocrine carcinoma (NEC), demonstrate marginal benefit with resection and are better managed with systemic therapies due to their aggressive behavior [10].

Even when significant liver clearance can be achieved, nearly all patients will have recurrence and residual disease [11]. Therefore, the goal of surgical resection is debulking to control symptoms and to slow disease progression. These goals can also be addressed with non-surgical treatment, and in addition to systemic therapies that include octreotide-based therapies with somatostatin receptor analogs, targeted therapies such as tyrosine kinase inhibitors and mToR pathway inhibitors, cytotoxic chemotherapy regimens such as streptozocin in combination with 5-FU or doxorubicin, and liver-directed therapies in the form of hepatic arterial embolization (HAE) and peptide receptor radionuclide therapy (PRRT) have demonstrated promising results [12,13,14]. This review will focus on the evaluation and surgical management of NETLMs, review non-surgical liver-directed therapies and PRRT, and propose a new classification system that divides NETLM patients into four categories with corresponding management strategies.

## 2. Modality of Choice for Optimal Detection of NETLM

As NETLMs often present as small, diffuse lesions throughout the liver, high recurrence rates up to 94% within five years after adequate hepatic debulking have been reported [15]. Therefore, while patients may derive survival benefit from cytoreduction, they are rarely considered cured. However, these high recurrence rates may be a consequence of poor radiographic detection and failure to clear unrecognized disease rather than recurrence itself as NETLMs are challenging to detect on imaging [16].

High-quality preoperative imaging for tumor localization and characterization of disease burden is critical for surgical planning and adequate hepatic debulking. A combination of multiple imaging modalities including triple-phase computed tomography (CT), magnetic resonance imaging (MRI), and gallium 68Ga DOTATATE positron emission tomography (PET) is utilized to detect NETLM. These tumors often display hyperenhancement on the arterial phase with persistent enhancement on the portal venous phase, but enhancement patterns can vary and smaller, sub-centimeter lesions may be missed [17]. It has been noted that 68Ga DOTATATE has an improved NETLM detection sensitivity of 94% compared to the reported sensitivities of 37.6% and 48.8% for CT and standard MRI, respectively [18]. However, the use of MRI with hepatocyte-specific contrast agents, such as gadoxetic acid (Eovist), has improved the sensitivity and specificity for NETLM detection over standard MRI and has been shown to impact surgical management [19]. The 20 min delay hepatobiliary phase with gadoxetic acid has been shown to demonstrate improved NETLM detection over both triple-phase CT and 68Ga DOTATATE [20,21,22]. A representative patient example showing the improved NETLM detection of the 20-phase delay hepatobiliary phase of MRI with Eovist over triple-phase CT, standard MRI with gadolinium, and 68Ga DOTATATE is shown in Figure 1.

In practice, all three modalities are supported by consensus guidelines for the evaluation of GEP-NET patients. While MRI with gadoxetic acid is the best modality to detect liver metastases, triple-phase CT is preferred for primary tumor detection and its anatomic relationship to neighboring structures such as blood vessels, and for assessment of lymph node metastases. Therefore, CT is typically the first modality obtained with consideration of MRI with gadoxetic acid when indeterminate liver lesions are seen. Moreover, 68Ga DOTATATE PET CT may allow for detection of the primary tumor (when not visualized with CT) and may also help delineate nodal or extrahepatic disease sites [23] to better define a patient’s extent of disease. Additionally, 68Ga DOTATATE PET CT can assess somatostatin receptor expression and predict response to PRRT [24]. Together, these modalities provide a comprehensive evaluation of GEP-NETs patients to optimize treatment strategies.

## 3. Hepatic Cytoreduction—Lowering the Debulking Threshold from 90% to 70%

The majority of NETLMs derive from either pancreatic (PNET) or small intestine (SI-NET) primaries, and up to 40% of PNETs are functional and secrete active hormones while 10% of SI-NET produce serotonin leading to carcinoid syndrome. Therefore, the initial indications for hepatic debulking were to mitigate hormonal symptomatology, and survival benefit was demonstrated in later studies. Additionally, the debulking threshold, or percentage of hepatic clearance that is required to derive clinical and survival benefit, has been examined over the past decades and has been lowered from 90% to 70% with good outcomes. Table 1 displays the pertinent studies that support the role of hepatic cytoreduction for NETLM and have defined the currently accepted debulking threshold. It should be noted that these studies are non-randomized, single-institution studies that compare their survival statistics against historical controls and therefore have potential inherent biases. However, as prospective trials have not been performed, these studies have provided the backbone for the currently accepted 70% debulking threshold.

The first case report of hepatic cytoreduction was published in 1977 by Foster and Berman and reported good syndrome control in 44 NETLM patients when at least 95% of their liver metastases were resected [25]. In 1990, McEntee et al. established the concept of a debulking threshold and reported that symptom control can be obtained for NETLM as long as at least 90% of liver disease is cleared [26]. Additional studies used this threshold and reported similar findings regarding symptom management with hepatic debulking [27].

In addition to improved symptomatology, hepatic cytoreduction has been associated with improved overall survival (OS) in appropriately selected patients with NETLM. In 2003, Sarmiento et al. was the first study to support this correlation [28]. In total, 170 patients with NETLM were included, and a 61% 5-year OS was reported using a debulking threshold of 90%, which was a significantly higher OS than in historical controls of reported survival for NETLM. Additionally, 96% of patients who underwent debulking experienced improvement in hormonal symptoms.

In 2008, Chamber et al. proposed that patients with NETLM may still obtain benefits with hepatic cytoreduction using a lower debulking threshold [29]. In a study with 66 patients with gastrointestinal NETs, 30 (45%) patients underwent hepatic cytoreduction. A 74% 5-year OS was reported, and hormonal symptoms were improved when at least 70% of liver disease was cleared. This debulking threshold of 70% was further supported by a study from Graff-Baker et al. that evaluated 52 patients with SI-NETs who underwent NETLM debulking [30]. Patients were divided into three groups based upon their degree of hepatic debulking (70–89% debulking, 90–99% debulking, and 100% debulking). The 5-year OS was 88% for the entire study, and there were no differences in liver progressive-free survival (PFS) or disease-specific survival between debulking groups, supporting lowering of the debulking threshold to 70%.

Maxwell et al. was the first study to demonstrate a survival benefit using a debulking threshold of 70% [31]. They described 108 patients with NETLMs who underwent hepatic cytoreduction ranging from less than 50% to greater than 90%. They found that patients with 70% or greater debulking had improved OS compared to patients with less than 70% debulking (median OS not reached versus 6.5 years; *p* < 0.05). An update to this study was published by the same institution by Scott et al. and reported outcomes from 188 patients that underwent hepatic cytoreduction [32]. OS was improved for >70% debulking compared to <70% debulking (median OS 134.3 vs. 37.6 months; *p* < 0.01). Although PFS was improved for patients that underwent >90% versus 70–90% debulking, there was no significant difference in OS between these groups. Another large, single institution study by Boudreaux et al. evaluated 189 with SI-NETs who underwent hepatic cytoreduction using a debulking threshold of 70% [33]. They reported 87% and 77% 5-year and 10-year OS, respectively, again supporting hepatic cytoreduction with a 70% debulking threshold after appropriate patient selection.

## 4. Role for Resection of the Primary Tumor with Or without Treatment of NETLM

While hepatic cytoreduction has been associated with symptom and survival benefit in patients with NETLM, the role of primary tumor resection (PTR) in these patients has been more controversial. For patients with SI-NET, PTR along with resection of its associated mesenteric mass reduces obstructive symptoms and abdominal pain [34]. The clinical and symptomatology benefits with PTR for patients with PNET is less clear. However, retrospective studies have demonstrated that PTR in the setting of NETLM, for both functional and non-functional tumors, is associated with improved overall survival. In a population-based study using the California Cancer Registry, 864 patients with GEP-NET and NETLM were identified. Patients that underwent PTR had improved OS compared to patients who did not, regardless of whether they received any treatment for their NETLM [35]. Patient who underwent both PTR and liver treatment had the longest survival. In another study of 2158 patients with non-functional PNETs from the SEER database, PTR resulted in prolonged overall survival in the entire cohort (median OS 1.2 vs. 8.4 years; *p* < 0.001) and for patients with metastases (median OS 1.0 vs. 4.8 years; *p* < 0.001) [36]. While these retrospective studies support PTR in patients with NETLM, it is important to note that they are population-based and may be limited by patient selection bias. However, PTR in patients with NETLM is recommended by consensus guidelines with appropriate patient selection and is standard practice at many centers.

**Table 1 jcm-12-02456-t001:** Studies defining the NETLM debulking threshold.

Year	Team	Number of Patients	Statistics	Debulking Threshold
1977	Foster and Berman [25]	44		95%
1990	McEntee et al. (Mayo) [26]	37	20-month OS: 83%	90%
1995	Que et al. (Mayo) [27]	74	4-year OS: 73%	90%
2003	Sarmiento et al. (Mayo) [28]	170	5-year OS: 61% 96% Symptom Improvement	90%
2008	Chamber et al. [29]	66	5-year OS: 74%	70%
2014	Graff-Baker et al. (OHSU) [30]	52	5-year OS: 88% No difference in PFS or DFS between debulking groups (70–89% vs. 90–99% vs. 100%)	70%
2014	Boudreaux et al. [33]	189	5-year OS: 87% 10-year OS: 77%	70%
2015	Maxwell et al. (Iowa) [31]	108	Improved Survival (Median OS NR for >70% vs. 6.5 months <70%; *p* < 0.05)	70%
2019	Scott et al. (Iowa) [32]	188	Improved Survival (Median OS 134.3 months >70% vs. 37.6 months <70%)	70%

## 5. Parenchymal-Sparing Techniques

Due to the high recurrence rate, hepatic resections with parenchymal-sparing approaches are favored to reduce morbidity and maximize remaining liver parenchyma for future hepatic debulking. As patient outcomes have been documented to be equivalent with R0 versus R1/R2 resections, hepatic wedge resections and enucleations are acceptable practices compared to formal hepatectomy [15,37]. Additionally, hepatic lesions up to 3 cm in size can be managed with ablation, and ablation can be combined with parenchymal-sparing surgical debulking to expand the numbers of patients who are candidates for cytoreduction [38].

The study by Maxwell et al. that demonstrated a survival benefit with cytoreduction using a debulking threshold of 70% supports this approach as a parenchymal-sparing approach (ablation, enucleation, and/or hepatic wedge resection) was used in the majority of patients [31]. Parenchymal-sparing techniques not only maintain opportunities for future resection and ablations but also maintain hepatic function for liver-direct therapies.

Regarding ablation, its two main indications are (1) as an intraoperative adjunct along with parenchymal-sparing surgical resection to meet the 70% debulking threshold and (2) for isolated, late recurrences that are better managed with a percutaneous approach rather than a repeat surgery. For example, a patient who recurs three years after their initial hepatic debulking with a single, small (<3 cm) left-sided liver lesion can undergo ablation rather than repeat partial hepatectomy.

Several ablation modalities are available and include laser ablation (LA), radiofrequency ablation (RFA), microwave ablation (MWA), and irreversible electroporation (IE). LA, RFA, and MWA cause tumor destruction through direct application of thermal energy, while IE utilizes nonthermal energy (high-voltage electrical current leading to generation of cellular membrane pores, loss of the physiologic ionic homeostasis, and ultimately cellular death) [39]. Currently, RFA and MWA are the most used modalities for both percutaneous and intraoperative ablation.

Comparing RFA and MWA, both can achieve large ablation zones. RFA can ablate up to approximately 3 cm with a single probe and up to 4–4.5 cm with up to three probes used simultaneously. MWA can provide a large ablation zone up to 4 cm with a single probe and 5–6 with threes probes used simultaneously. In addition to its short time interval of 5–10 min, MWA is superior to RFA, in that it is not as limited by the heat-sink effect [40]. Therefore, while both RFA and MWA are still widely used, many centers now favor the use of MWA over RFA due to the faster heat times, larger ablation zone, and lower heat-sink effect seen with MWA [41].

## 6. A New Classification System to Guide Surgical Management of NETLM

Although surgeons should strive for complete resection and clearance of disease whenever possible, in practice this is frequently not achievable for NETLM as patients commonly present with diffuse, bilobar disease, and small lesions are challenging to detect. Therefore, both the lowering of the debulking threshold to 70% and the use of parenchymal-sparing surgical techniques and ablation have greatly expanded the number of NETLM patients that are potential candidates for hepatic cytoreduction. Appropriate patient selection is critical to ensure that only patients who will derive benefit undergo surgery. Therefore, we propose a new classification system and treatment algorithm for NETLM to optimize patient selection for hepatic cytoreduction.

As shown in Figure 2, this classification system divides patients into four categories based upon the burden of their hepatic disease and ability to meet the 70% debulking threshold with cytoreduction. The categories are as follows:Type 1: Patients of this category have a limited number of NETLMs that can be completely cleared with hepatic debulking. Depending on the degree of hepatic resection, synchronous PTR can be performed at the time of hepatic cytoreduction as per surgeon discretion.Type 2: Patients have multiple lesions diffusely throughout the liver, and >70% debulking can be achieved utilizing parenchymal-sparing techniques and ablation. PTR should be considered and can be performed synchronously or as a separate procedure depending on the extent of operation required.Type 3: Patients have extensive, bilobar hepatic involvement, but unlike Type 2 patients, >70% debulking clearance cannot be achieved and cytoreduction should not be performed. These patients are better candidates for liver-directed therapies such as HAE with radioembolization. However, they should be evaluated for PTR as survival benefit is demonstrated even without liver-directed interventions as discussed previously in this review.Type 4: Patients have extensive hepatic involvement that is profoundly symptomatic from either the mass effect from large, bulky tumors (often from impending venous occlusion due to tumor compression of the IVC, hepatic veins, or portal vein), or from hormonal symptomatology that cannot be medically mitigated. Although >70% debulking cannot be achieved, select patients may have improved quality of life with palliative hepatic debulking of these symptomatic lesions. As survival is unlikely to be substantially prolonged, palliative cytoreduction must be carefully weighed with potential surgical morbidity.

Other considerations must be applied when evaluating NETLM patients for hepatic debulking. Contraindications to hepatic cytoreduction include poor general health and inability to tolerate surgery (including carcinoid heart disease), and extrahepatic tumor metastases such as peritoneal carcinomatosis. While the use of hepatic cytoreduction and liver directed therapies for NETLM with limited extrahepatic disease is being examined, is it not recommended under current consensus guidelines. Disease biology and progression rates must also be considered; surgical debulking is less likely to benefit patients with rapidly progressing or high-grade disease due to the high likelihood of early recurrence. Therefore, poorly differentiated tumor histology (classified as NEC) is typically another contraindication for surgery.

Additional considerations include that debulking should only be performed when an adequate liver remnant with preserved hepatic function is left behind. A thorough evaluation of baseline hepatic function must be performed. Preoperative portal vein embolization can be utilized in patients with a small predicted future liver remnant (FLR) to expand their candidacy. Regarding PTR, it is our institution’s practice to perform synchronous PTR at the time of hepatic cytoreduction, and this is achieved in over 90% of both Type 1 and 2 patients. Pancreaticoduodenectomy should only be performed in select patients due to increased surgical morbidity and the potential for hepatic arterial compromise leading to a higher risk of liver abscesses [42]. Lastly, per current consensus guidelines, prophylactic cholecystectomy should be performed at the time of debulking surgery as the patient is likely to require long-term somatostatin analogs and is at higher risk for gallstone formation.

Following these surgical considerations and use of our proposed classification system can provide guidance as to how to manage patients with NETLM. The majority of these patients will fall into the Type 2 or Type 3 categories, and it is therefore critical to determine whether a 70% or higher cytoreduction with parenchymal-sparing debulking and ablation can be performed. Type 2 patients that meet this threshold should be evaluated for surgery, while Type 3 patients are better served with liver-directed therapies or PRRT. Type 2 patients that are poor surgical candidates or have an extrahepatic disease should also be managed like Type 3 patients. Careful patient selection must be performed for all patients prior to rushing to the operating room as the promising results and expanding use of liver-direct therapies and PRRT provide a valuable alternative treatmentfor NETLM patients. Therefore, the next sections will review these non-surgical treatment modalities.

## 7. Liver-Directed Therapies

Outside of systemic therapies, which include octreotide-based therapies, targeted therapies, and cytotoxic chemotherapy, liver-directed therapies are being more commonly utilized for unresectable hepatic disease for both tumor control and symptom management. Good candidates for these therapies include patients that have a high volume of liver disease, progressive disease, or symptomatic liver disease that is not resectable. These therapies are based on the blood supply of NETLM; while normal liver parenchyma is primarily supplied by inflow from the portal vein, NETLMs derive their blood supply almost exclusively from the hepatic arterial system [43]. Therefore, hepatic arterial embolization (HAE) is effective for both somatostatin-positive and somatostatin-negative tumors (unlike somatostatin analogs and PRRT), and it can target NETLMs without impacting the surrounding normal liver parenchyma.

There are three commonly used techniques for HAE that include transarterial bland embolization (TAE) with gel foam, transarterial chemoembolization (TACE) with gel foam combined with chemotherapy or drug-eluting beads, and transarterial radioembolization (TARE) with yttrium-90 (Y90). [44]. TAE entails catheterization of the hepatic artery and embolization using calibrate microparticles of ethiodized oil associated with gelform [45]. With TACE, embolization is performed with cytotoxic chemotherapy reagents such as cisplatin, doxorubicin, gemcitabine, or mitomycin C. These chemotherapy reagents can be delivered as an oily emulsion or as drug-loaded beads. TACE is thought to be effective for NETLM as the ischemia achieved with embolization results in the tumor cells being more suspectable to chemotherapy. While TAE and TACE are generally well-tolerated, major complications, including liver failure, ischemic cholecystitis, acute carcinoid syndrome, and liver abscess, have been reported in up to 17% of patients [46]. However, a substantial improvement in both symptom control (50–90%) and tumor response has been demonstrated with a hepatic PFS of over 10 months [47].

TARE with Y90, also referred to as selective internal radiation therapy (SIRT), consists of catheterization of the hepatic arterial system with embolization using radioisotopes (most commonly Y90) carried by embolization agents (resin or glass spheres) [48]. In addition to inducing ischemia through embolization, this approach treats NETLM by selectively directing radiation therapy. SIRT was approved by the Food and Drug Administration (FDA) in 1999 for the management of metastatic liver or primary liver malignancies with limited systemic treatment options. A meta-analysis by Jia et al. examined 870 patients with unresectable NETLM who underwent embolization with Y90 [49]. The median disease control rate was 86% at 3 months post-treatment, and median OS was 28 months with a 1-, 2-, and 3-year OS of 72.5%, 57%, and 45%, respectively. In total, 69.1% of patients with hormonally symptomatic disease demonstrated improvement in their symptoms. Overall, this therapy was well-tolerated. Major complications were rare, but included radiation gastritis, duodenal ulcers, radiation cholecystitis, and death due to liver failure. The efficacy and safety of SIRT have been demonstrated by additional studies [50,51].

All three HAE approaches are effective and well-tolerated, and they can be repeated every 4–6 weeks and used in combination (SIRT can be performed after unsuccessful TAE or TACE). To improve tolerance and safety, it is common practice to only treat one hepatic lobe per session. They also have been demonstrated to be safe after hepatic debulking [52], and limited extrahepatic disease is not a contraindication for their use [43].

In addition to its indication for palliation of unresectable hepatic disease, SIRT has been used as a bridge to surgical resection in other malignancies such as metastatic hepatocellular carcinoma or colorectal cancer [53,54]. SIRT has also been examined as a strategy to induce liver hypertrophy in the contralateral FLR [55]. The role of neoadjuvant SIRT for unresectable NETLM is less defined. In a retrospective, multicenter study of patients with unresectable liver metastases, the feasibility and safety of hepatectomy after neoadjuvant SIRT was examined [56]. A total of 47 patients were included, with the majority having hepatocellular carcinoma (30%) or colorectal liver metastases (23%). In total, 8 (17%) of the patients had NETLM. Sixty-four percent of patients underwent right or extended right hepatectomies, and 10 had simultaneous ablations. Thirty-one (66%) patients had pathologically confirmed R0 resections. Regarding complications, 27.7% (13/47) had a Grade 3a or higher complication as per the Clavien–Dindo classification system, one patient had post-hepatectomy liver failure (he ultimately recovered), and there was one death from intraoperative hemorrhage. At a median follow-up of 40 months, 70% of the patients were either without disease recurrence or alive with disease. This study concluded that hepatectomy after SIRT is safe and results in acceptable survival. Additional studies are warranted to define the role of neoadjuvant liver-directed therapies in the management of NETLM.

An additional liver-directed therapy that utilizes the hepatic arterial system for drug delivery is hepatic artery infusion (HAI). While this approach is becoming more commonly utilized in the management of metastatic colorectal cancer and cholangiocarcinoma, it is rarely used for NETLM. In one study, 77 patients with NETLM underwent HAI followed by TACE or HAI alone [57]. The overall response rate (radiographic response or symptom control) was 80%. Median disease-specific survival was 39 months, which was not superior from the outcomes of other forms of HAE without HAI. Therefore, HAI for NETLM is still not standardly performed, and additional studies are required to support its utility.

## 8. Peptide Receptor Radionuclide Therapy

For patients with refractory metastatic NET, peptide receptor radionuclide therapy (PRRT) has shown promising results. Lutetium-177 (177Lu) dotatate, which goes by the commercial name Lutathera, was approved by the FDA in 2018 for the treatment of somatostatin-receptor positive GEP-NETs. This approval was based on results from the phase III, multicenter, open-label, randomized controlled trial NETTER-1 [58]. In this trial, 229 patients with progressive, well-differentiated, inoperable or metastatic somatostatin receptor-positive midgut NETs were randomized to receive 177Lu-Dotatate with long-acting octreotide or octreotide alone. The 177Lu-Dotatate had significantly longer PFS compared to the control group (65.2% vs. 10.8% at 20 months). The trial’s long-term results regarding OS were reported in 2021, and although not significant, the 177Lu-Dotatate group had a 11.7 longer median OS compared to the control group [59]. PRRT is therefore an important treatment option in the multimodal approach for the management of NETLM.

Additionally, PRRT has been shown to result in downstaging of unresectable primary tumors allowing for resection in select patients. Therefore, it has been studied in the neoadjuvant setting for initially unresectable somatostatin receptor-positive primary GEPNETs. In one study, 23 patients with resectable or potentially resectable PNETs at high risk for recurrence who underwent neoadjuvant PRRT followed by surgery were compared with 23 matched patients who underwent upfront surgical operation [60]. While neither the median disease-specific survival nor PFS differed between groups, PFS in the 31 patients with a R0 resection was greater in the 15 patients who underwent neoadjuvant PRRT compared to the 16 patients in the upfront surgery group (median PFS not reached vs. 36 months; *p* < 0.05). Pancreatic fistula was also found to be lower in the PRRT group vs. the upfront surgery group (0/23 vs. 4/23; *p* < 0.02). Studies have also looked at the rate of conversion from unresectable to resectable primary tumors after neoadjuvant PRRT [61]. One study examined 57 GEP-NET patients with unresectable primary tumors due to vascular involvement who received PRRT [62]. Of the study patients, 23 were without liver metastases and 34 had potentially resectable liver metastases. After PRRT, 26.3% (15/57) of patients were converted to resectable, which included 7 patients without liver metastases and 8 patients with liver metastases. Conversion of the primary tumor to resectable was more likely for tumors with the following characteristics: small size (<5 cm), duodenal origin, absence of regional lymph node involvement, lower intensity of 18F-FDG uptake, and fewer and small liver metastases. The 2-year OS for all patients was 92.1% (95% without and 90% with liver metastases), supporting neoadjuvant PRRT as a potential option for GEP-NET with initially unresectable primary tumors.

## 9. Liver Transplantation

The role of liver transplantation (LT) for patients with unresectable NETLM without an extrahepatic disease remains controversial due to numerous factors including high rates of recurrence after LT [63] and the indolent nature of NETLM resulting in long-term survival. In the largest cohort study to date, Le Treut et al. reported on 213 patients with NETLM who underwent LT throughout a 27-year time interval from the European Liver Transplant Registry [64]. In this study, 5-year OS was reported at 52%, and recurrence occurred in 60% of patients. In a meta-analysis including 279 patients with NETLM who underwent LT, 1-, 3-, and 5-year survival rates were 89%, 69%, and 63% [65]. However, recurrences after LT were frequent, and the rates ranged from 31.3% to 56.8%. Therefore, although LT is an acceptable treatment as per OPTN/UNOS guidelines for unresectable NETLM without EHD and as long as the primary tumor has been resected, it is not routinely performed. Of the total LTs performed in the United States and Europe, only 0.2–0.3% are performed for NETLM [66]. Larger studies are needed to validate LT for NETLM, and LT may be a good option for select Type 3 and 4 NETLM patients that cannot achieve the debulking threshold with surgery.

## 10. Conclusions

GEP-NETs are associated with a high rate of liver metastases, and although not curative, cytoreduction in the form of hepatic parenchymal-sparing debulking has been associated with improved symptom control and prolonged survival. It is suggested that 68Ga DOTATATE and gadoxetic acid-enhanced liver MRI are the best imaging modalities to preoperatively characterize the extent of NETLM, and PTR is associated with improved survival regardless of whether liver-directed therapies or hepatic debulking are performed. Our proposed classification system divides patients into four groups based upon hepatic disease burden and symptoms, and it provides guidance for the surgical management of NETLM. Outside of surgery and systemic therapies, liver-directed therapies (HAE) and PPRT are emerging treatment options with promising results. Due to the large number of NET treatment options now available, all patients should be evaluated by a multidisciplinary team of experts to optimize their care. A multimodal approach has transformed the natural history of well-differentiated metastatic NET to a slowly progressive chronic illness where longevity with good quality of life has become the standard of care.

## Figures and Tables

**Figure 1 jcm-12-02456-f001:**
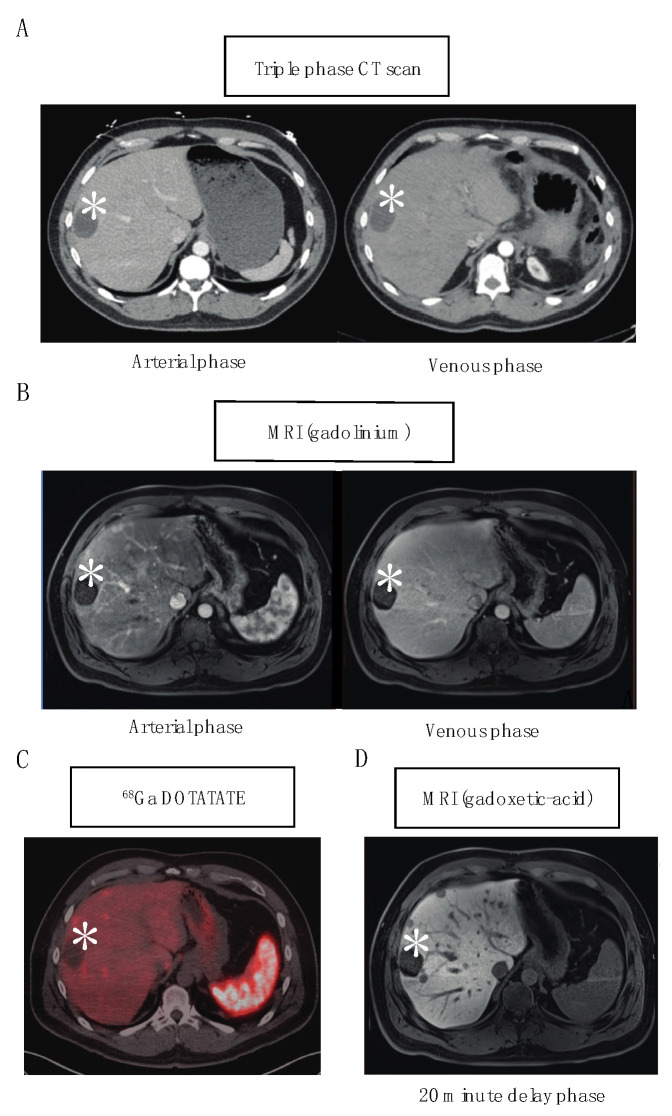
Representative example demonstrating NETLM detection for one patient with different imaging modalities including (**A**) triple-phase CT scan (arterial and venous phases) (**B**) standard MRI with gadolinium contrast (arterial and venous phases) (**C**) 68Ga DOTATATE PET CT imaging (fused phase) and (**D**) MRI with hepatocyte-specific contrast agent gadoxetic acid (20 min delay phase). * symbol denotes hepatic cyst demonstrating that each representative image corresponds to the same anatomic location.

**Figure 2 jcm-12-02456-f002:**
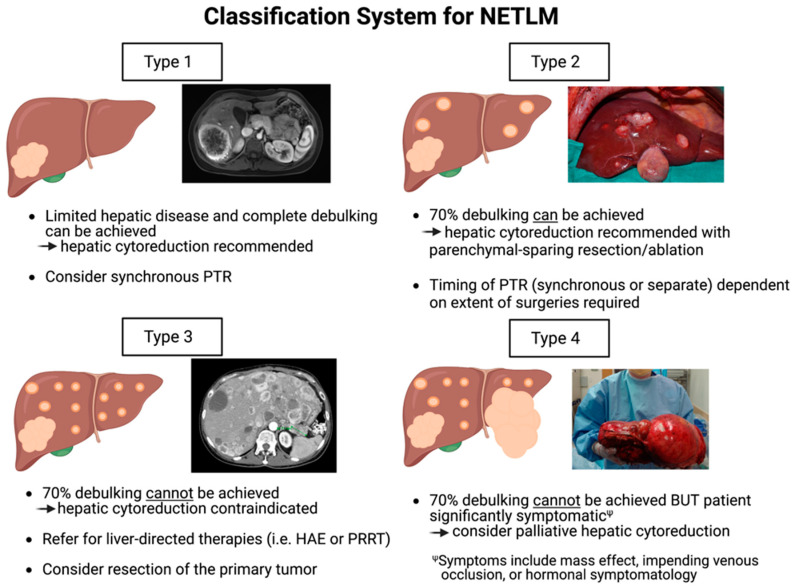
Proposed classification system of neuroendocrine tumor liver metastases (NETLM) to guide surgical management. The four types are defined by hepatic involvement, ability to achieve the recommended 70% debulking threshold with cytoreduction, and symptoms.

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
