# Peer review of "Defining a New Classification System for the Surgical Management of Neuroendocrine Tumor Liver Metastases"

_jcm, 2023, doi:10.3390/jcm12072456_

Round 1

Reviewer 1 Report

This is a well-written review of the surgical management of neuroendocrine tumor liver metastases. The authors have suggested a new classification system based on the extent of hepatic disease, potential for liver debulking, and primary tumor resection. 

I have a few minor suggestions:

1. Line 140: should state 70%

2. Line 236: change recommendation to recommended 

3. Line 238: change to thorough

4. Line 253: change to "while Type 3 patients that do not are better served with liver-directed 253 therapies or PRRT"

5. Line 310: Change colorectal cancer to 'colorectal liver metastases

6. Line 312: Should state as 31 (66%) of patients 

Author Response

We appreciate the reviewer's comprehensive review and encouraging feedback. We agree with all suggested changes and have made the corrections.

1. Line 140: should state 70%.

We agree and this change was made.

2. Line 236: change recommendation to recommended 

We agree and this change was made. 

3. Line 238: change to thorough

We agree and this change was made. 

4. Line 253: change to "while Type 3 patients that do not are better served with liver-directed 253 therapies or PRRT"

We agree and this change was made. 

5. Line 310: Change colorectal cancer to 'colorectal liver metastases

We agree and this change was made. 

6. Line 312: Should state as 31 (66%) of patients 

We agree and this change was made. 

Reviewer 2 Report

The authors present a comprehensive and excellent review of the current state of surgical management of neuroendocrine liver metastases. The proposed classification system for the types of NET liver metastases is novel. Would consider adding consideration for liver transplantation for types 3 and 4 and also discuss the UNOS criteria for liver transplant in the appropriate section.

Author Response

Point 1: The authors present a comprehensive and excellent review of the current state of surgical management of neuroendocrine liver metastases. The proposed classification system for the types of NET liver metastases is novel. Would consider adding consideration for liver transplantation for types 3 and 4 and also discuss the UNOS criteria for liver transplant in the appropriate section.

Point 1 Reply: We appreciate the reviewer’s encouraging feedback. In response to the suggestion, we expanded upon the liver transplantation section and included the criteria for liver transplantation for NET. We also included it's potential as a treatment option for the Type 3 and 4 patients we propose in our classification. 

Reviewer 3 Report

Review discussing treatment options vor livermetastases of neuro-endocrine tumors. Update of new techniques and studies published concerning these techniques.

Overall the manuscript does not really include some appriciation of study limitations. Also the manuscript does not show some critical insight in patient treatment effects. For example the treatment planning for th eliver does not include the disease progression rate and symptom relief is no longer a goal of debulking liver metastases.

The manuscript promotes liver treatment but should have included more critical notes and provided also some insight in palliative and wait and see policies in these patients.

Specific remarks;

Introduction

-          Gold standard for NETLM is surgery? Satatement should be more balanced

-          Debulking nowadays less necessary since symptom-relieve is often achieved with medication, to slow disease progression is dubious especially in the light of complications; goal of surgery should be further explained, more in the light of complete tumor resections.

Debulking paragraph / cytoreduction

-          Evidence not properly discussed; inclusion o fpatinets was not randomized; only selected patients were included in these studies and compared to historic controls or other debulking rates with no explanation for operative decision making

Primary tumor resection

-      Again no limitations mentioned; population based studies possibly only included the fit patients for surgery of the primary tumor.

Discussion of four types of patients should include disease progression rates since progressive disease patients most likely will deteriorate from liver directed surgery instead of gaining.

Cholecystectomy Is not obligatory.

The conclusions are a bit short and should have included for example the importance of an expert clinic in the treatment of NET/NEN and should included, according to NANETS and ENETS guidelines the multi disiciplinary board were discussions on treatment should take place and a balanced disicion on treatment given.

The 4 group sof liver metastases are not new and are not improving the discussion on liver therepy.

Author Response

We appreciate the reviewer's comprehensive review and feedback and have addressed each comment as discussed below.

Point 1: Gold standard for NETLM is surgery? Satatement should be more balanced

Point 1 reply: We have correct the line to state that the "The gold standard for curative intent management of NETLM is surgery" as we agree that treatment of NETLM can include many treatment modalities. However, it is a commonly accepted view that curative intent treatment of NETLM requires that surgical resection be a component of therapy.

Point 2: Debulking nowadays less necessary since symptom-relieve is often achieved with medication, to slow disease progression is dubious especially in the light of complications; goal of surgery should be further explained, more in the light of complete tumor resections.

Point 2 reply: We have further explained that while symptom relief and slowing disease progression is typically the goal of surgical debulking, non-surgical therapies can also achieve these goals.

Point 3: Evidence not properly discussed; inclusion o fpatinets was not randomized; only selected patients were included in these studies and compared to historic controls or other debulking rates with no explanation for operative decision making

Point 3 reply: As suggested by the reviewer, we have addressed the limitations of these studies as they are retrospective and single-institution.

Point 4: Again no limitations mentioned; population based studies possibly only included the fit patients for surgery of the primary tumor.

Point 4 reply: As suggested by the reviewer, we have included that the discussed studies have inherent selection bias as they are retrospective studies.

Point 5: Discussion of four types of patients should include disease progression rates since progressive disease patients most likely will deteriorate from liver directed surgery instead of gaining.

Point 5 reply: We have included in our discussion of the four types of NETLM patients defined by our classification system that disease progression is a critical factor when determining if a patient should undergo hepatic disease debulking.

Point 6: Cholecystectomy Is not obligatory.

Point 6 reply: While we appreciate the reviewer's point that some patients with NETLM may not require cholecystectomy, we continue to keep it as a recommendation as current consensus guidelines (i.e. NANETs) recommend cholecystectomy at the same of hepatic debulking for all patients with NETLM if feasible due to their high-likelihood of requiring somatostatin analogues and therefore high-risk of developing gallstones. While this practice pattern may not be utilized by all institutions, it is the current national guidelines. 

Point 7: The conclusions are a bit short and should have included for example the importance of an expert clinic in the treatment of NET/NEN and should included, according to NANETS and ENETS guidelines the multi disiciplinary board were discussions on treatment should take place and a balanced disicion on treatment given.

Point 7 reply: We appreciate the reviewer's comment and have added the importance of mutidisciplinary review by an expert team of NET specialists to determine the optimal multimodal approach.

Point 8: The 4 group sof liver metastases are not new and are not improving the discussion on liver therepy.

Point 8 reply: We appreciate the reviewer’s point of view, but with the utmost respect we believe our classification system is novel and helps provide a framework for how to think about selecting patients for liver resection or other therapeutic approaches. We believe our classification system  constitutes a significant contribution to NETLM evaluation and management. 

Round 2

Reviewer 3 Report

The manuscript is improved and many remarks have been resolved.

I personally still feel that the effect of surgical treatment NELM is overrated and the data of previous studies comes from selected patient groups. It is a strong believe of the authors that a partial /debulking operation has survival benefits,. They believe even more benefits than systematic treatment options.

I do not share this thought and this manuscript will be used by surgeons to support a treatment that is not without harm in an often frail population.

Possibly a more reflective or more nuanced discussion would be better, 

It would be better to promote NELM resection with the intent to perform complete resections and to advice against debulking surgery unless symptoms are no longer under control with systemic treatment.

Author Response

We appreciate the reviewer's feedback regarding our manuscript and have taken the comments into consideration.

We agree that non-surgical treatment options are a critical element of care for NETLM and that all patients should be evaluated by a multidisciplinary team. Many patients with NETLM with be better served with liver-directed therapies or PRRT rather than hepatic debulking, and we have added additional sentences in our section regarding our classification system to emphasize this. As suggested by the reviewer, we include that complete surgical resection should be the main intent for surgery, however, it is often not feasible in the reality of practice. We are often included that careful patient is critical and patients shouldn't be rushed to the operating room as non-surgical therapies may provide good outcomes.

However, with the utmost respect we do stand by our statements the liver debulking and primary tumor resection can provide survival benefit with appropriate patient selection.  Although it is true these studies had a retrospective design, they are accepted by many providers and institutions as standard of care and form the backbone of many of the current NANETS recommendations. Two of the major causes of death of NETLM patients are 1) hepatic failure from disease replacement or 2) mesenteric sclerosis from the primary tumor, and so surgical intervention that addresses these issues may provide benefit to appropriate patient.  As the focus of this chapter is the surgical management of NETLM, we feel it appropriate to highlight the role of hepatic debulking as a component of a successful multimodal approach to NETLM management.